# Role of Cardiac Magnetic Resonance Imaging in the Evaluation of Athletes with Premature Ventricular Beats

**DOI:** 10.3390/jcm11020426

**Published:** 2022-01-14

**Authors:** Giulia Brunetti, Alberto Cipriani, Martina Perazzolo Marra, Manuel De Lazzari, Barbara Bauce, Chiara Calore, Ilaria Rigato, Francesca Graziano, Riccardo Vio, Domenico Corrado, Alessandro Zorzi

**Affiliations:** Department of Cardiac, Thoracic and Vascular Sciences and Public Health, University of Padova, Via Giustiniani 2, 35128 Padova, Italy; giulia.brunetti@studenti.unipd.it (G.B.); alberto.cipriani@unipd.it (A.C.); martina.perazzolomarra@unipd.it (M.P.M.); manuel.delazzari@aopd.veneto.it (M.D.L.); barbara.bauce@unipd.it (B.B.); chiara.calore@aopd.veneto.it (C.C.); ilaria.rigato@aopd.veneto.it (I.R.); francesca.graziano.1@studenti.unipd.it (F.G.); riccardo.vio@studenti.unipd.it (R.V.); alessandro.zorzi@unipd.it (A.Z.)

**Keywords:** cardiac magnetic resonance, ventricular arrhythmias, sudden cardiac death, sports cardiology, athletes, preparticipation screening

## Abstract

Premature ventricular beats (PVBs) in athletes are not rare. The risk of PVBs depends on the presence of an underlying pathological myocardial substrate predisposing the subject to sudden cardiac death. The standard diagnostic work-up of athletes with PVBs includes an examination of family and personal history, resting electrocardiogram (ECG), 24 h ambulatory ECG (possibly with a 12-lead configuration and including a training session), maximal exercise testing and echocardiography. Despite its fundamental role in the diagnostic assessment of athletes with PVBs, echocardiography has very limited sensitivity in detecting the presence of non-ischemic left ventricular scars, which can be revealed only through more in-depth studies, particularly with the use of contrast-enhanced cardiac magnetic resonance (CMR) imaging. The morphology, complexity and exercise inducibility of PVBs can help estimate the probability of an underlying heart disease. Based on these features, CMR imaging may be indicated even when echocardiography is normal. This review focuses on interpreting PVBs, and on the indication and role of CMR imaging in the diagnostic evaluation of athletes, with a special focus on non-ischemic left ventricular scars that are an emerging substrate of cardiac arrest during sport.

## 1. Introduction

Adolescents and young adults engaged in competitive sports are at increased risk of sudden cardiac death (SCD) compared with their sedentary counterparts. Sport activity can act as a trigger for life-threatening ventricular arrhythmias (VAs) in athletes with an underlying cardiovascular disease, predisposing them to adrenergic-induced ventricular fibrillation [1,2,3,4]. Some authors have hypothesized that intense exercise may aggravate underlying pathological myocardial substrates or even cause an exercise-induced cardiomyopathy [5,6,7].

Documentation of premature ventricular beats (PVBs) on 12-lead electrocardiography (ECG), ambulatory ECG monitoring or ECG stress tests should raise the suspicion of an underlying cardiomyopathy. PVBs with common morphologies such as infundibular or fascicular ectopic beats are often not associated with structural cardiac abnormalities, whereas other PVBs may represent a red flag for the presence of a pathological substrate and must be properly investigated [8,9]. Standard examinations, including echocardiography, may fail to detect structural arrhythmic substrates such as a non-ischemic left ventricular scar (NLVS), which can only be detected by contrast-enhanced cardiac magnetic resonance (CMR) imaging.

This review focuses on interpreting PVBs, and on the indication and role of CMR imaging in the diagnostic evaluation of athletes, with a special focus on non-ischemic left ventricular scars that are an emerging substrate of cardiac arrest during sport.

## 2. Substrates of PVBs in Athletes

Malignant arrhythmic events may occur in athletes with a structurally normal heart as a result of genetic channelopathies [10], some of which can be suspected based on ECG abnormalities (e.g., long and short QT syndromes and J wave syndromes), while others need exercise testing to be unmasked (e.g., catecholaminergic polymorphic ventricular tachycardia). The latter condition is a genetic ion channel disease leading to increased calcium release from the sarcoplasmic reticulum following adrenergic stimulation [11]. The disease should be suspected if exercise elicits PVBs with multiple morphologies (both RBBB- and LBBB-like) that worsen in number and complexity with increasing effort [12].

In master athletes (age >35 years), the most important arrhythmic substrate is ischemic heart disease. Male master endurance athletes, even with a low atherosclerotic risk profile, are more likely to have a higher CAC score compared with sedentary males with a similar risk profile [13]; the meaning of this observation is still uncertain. Exercise testing in athletes with coronary artery stenosis may elicit short-coupled PVBs, with an “R on T” phenomenon, which may trigger runs of non-sustained polymorphic ventricular tachycardia (VT). In our experience, this peculiar arrhythmic pattern may be the only sign of underlying myocardial ischemia even in the absence of ST segment depression or chest pain.

Beyond channelopathies and ischemic heart disease, different structural cardiomyopathies can be associated with PVBs, including genetic and acquired heart muscle diseases. Hypertrophic cardiomyopathy (HCM) has been reported to cause more than one third of SCDs in the USA [14,15], while arrhythmogenic cardiomyopathy (ACM) accounts for approximately one fourth of the cases in the Veneto region of Italy [1,3,16]. In most cases, first (history, physical examination, ECG) and second-level testing (stress test ECG, ambulatory ECG and echocardiography) is enough to exclude an underlying cardiomyopathy; however, some concealed arrhythmic substrates can be missed unless advanced imaging tests are used.

NLVSs have been associated with cardiac arrest occurring during effort [17,18]. In 2016, Di Gioia et al. reported a postmortem series of young SCD victims and found that NLVSs accounted for 25% of sport-related fatalities, while they were uncommon substrates in events unrelated to sports [19]. Interestingly, the scar was often associated with histopathologic evidence of fibrofatty replacement of the right ventricular myocardium, suggesting that the NLVS may be the expression of a biventricular ACM rather than the result of prior myocarditis. Biventricular myocardial fibrosis was found in the postmortem examination of a famous Italian footballer who died suddenly during a competition [20]. No matter the cause, a NLVS should be regarded as clinically significant because it is a possible substrate for macro-reentrant VT [21,22,23,24].

Non-ischemic myocardial fibrosis, in contrast to the ischemic one, involves the mid-mural and subepicardial myocardial layers but spares the endocardium. Since the latter contributes most to myocardial contractility, echocardiographic wall motion evaluation has limited sensitivity in detecting the presence of a NLVS [25]. ECG abnormalities such as low QRS voltages in the limb leads (“peak-to-peak” QRS amplitude <0.5 mV) and/or T wave inversion in the left precordial leads are associated with an underlying NLVS on CMR, even if their sensitivity does not exceed 20–30%. The association between low QRS voltages and LV scars has been explained by the replacement of the LV myocardial mass with electrically inert fibrous tissue [26].

## 3. Role of CMR in the Diagnostic Work-Up of Athletes with PVBs

Cardiac magnetic resonance is the gold standard for the assessment of biventricular volume, mass and global and regional function [27,28]; LV function can be quantified using simplified CMR approaches such as biplanar long-axis views or short-axis stacks [29], and normal reference values for biventricular size and function have been published for the general population [30] and for athletes [31]. In addition to volume and function assessment, cine-CMR allows a more confident evaluation of regional abnormalities (such as asymmetric hypertrophy, hallmarks of HCM or the presence of crypts, diverticula or aneurysms); moreover, multiplanar cine-CMR can be used to assess valve morphology and function, and the aortic root with the evaluation of the origin and course of coronary arteries [32].

Recently, myocardial strain analysis on feature-tracking CMR, following the footsteps of speckle-tracking echocardiography, has emerged as a surrogate marker of LV systolic function, analyzing the deformation of a fixed myocardial point through the cardiac cycle; it is of prognostic value in different cardiomyopathies such as HCM, dilated cardiomyopathy, ischemic cardiomyopathy or myocarditis [33,34,35,36,37,38].

Beyond the role of functional indices, CMR offers the unique ability to identify myocardial tissue abnormalities that may have been missed by other imaging techniques. Tissue abnormalities typical of cardiomyopathies reflect chronic injury, inflammation, infiltration or abnormal storage that can be disclosed by specific CMR sequences. Standard CMR techniques for tissue characterization include T2-weighted CMR images, with an increased signal intensity reflecting inflammation-related myocardial edema [39,40]; T1-weighted black-blood images, where fat appears as a hyperintense bright signal; and contrast-enhanced T1-weighted scans performed after intravenous administration of gadolinium-based contrast agents, which has emerged as a widely available technique to enable visualization and quantification of myocardial fibrous tissue.

Over the last two decades, it has been demonstrated that a higher scar volume quantified with late gadolinium enhancement (LGE) predicts the risk of life-threatening VA recurrence in patients with aborted SCD [41], and the complexity of VAs in patients with non-ischemic cardiomyopathy referred for ablation of PVBs [42]. A recent meta-analysis of the impact of LGE on various cardiovascular outcomes both in ischemic and non-ischemic cardiomyopathies concluded that the presence of LGE was strongly associated with SCD and VAs [21]. The presence of LGE with a subepicardial/midmyocardial distribution without the diagnostic features of a specific cardiac disease is defined as an “isolated NLVS” (Figure 1). Although an isolated NLVS is traditionally interpreted as the consequence of previous myocarditis [43], other diseases such as the left-dominant ACM or cardiac involvement in sarcoidosis may manifest with this LGE pattern on CMR [44,45,46,47,48]. Some studies have reported a higher prevalence of NLVSs in endurance athletes compared to controls, suggesting that high-intensity exercise may cause cardiac fibrosis [49].

Myocardial mapping, a technique that has been recently introduced in clinical practice, allows measuring the different magnetic properties of each myocardial volume as expressed by magnetic relaxation times (native T1, T2, T2*) and the extracellular volume (ECV) derived from post-contrast T1. Changes in magnetic properties are typical of diseases characterized by intracellular cardiomyocyte disturbances (e.g., iron overload, or glycosphingolipid storage in Anderson–Fabry disease), extracellular disturbances in the myocardial interstitial tissue (e.g., myocardial fibrosis or amyloid deposit) or both (e.g., myocardial edema and/or infarction with increased intracellular and extracellular water) [50,51]. Conventional T1- or T2-weighted images require a subjective assessment, while mapping allows the quantification of the disease process. Differentiating cardiomyopathies from the physiologic adaptation of the athlete’s heart is one of the emerging potential clinical uses of mapping techniques [52,53,54,55]. Since T1 mapping represents a novel marker of myocardial fibrosis, it may complement LGE in the evaluation of athletes with VAs, although scientific evidence is still lacking (Figure 2).

Finally, arrhythmogenic substrates characterized by inflammation, such as cardiac sarcoidosis, may be better disclosed by integrating CMR with nuclear imaging, particularly positron emission tomography (PET) [56,57]. Cardiac involvement in sarcoidosis can lead to inflammation and subsequent fibrosis that, on CMR, is characterized by multiple foci of LGE that may mimic ACM. In this subset, a recent study by Muser et al. demonstrated the usefulness of PET imaging, as compared to contrast-enhanced CMR, in accurately characterizing electroanatomic abnormalities responsible for VAs and potential targets for substrate ablation [58].

## 4. Modern Interpretation of Premature Ventricular Beats in Athletes

According to the current criteria for ECG interpretation in athletes, abnormalities are labeled as either “common” or “uncommon” [59,60]. The early repolarization pattern, for example, is included among the common abnormalities because it is related to training and found in a wide proportion of athletes because of the electrical remodeling of the heart in response to training. Conversely, T wave inversion in lateral leads is considered uncommon and unrelated to training because it is infrequently found in healthy athletes and thus requires an appropriate clinical work-up to exclude an underlying disease.

The modern classification of PVBs follows a similar approach: “common” PVBs are usually idiopathic (i.e., with a structurally normal heart), whereas “uncommon” PVBs carry a higher probability of an underlying myocardial disease (Table 1).

PVB morphology is the most relevant feature. Athletes with frequent PVBs in the absence of cardiac disease most often show a morphology suggesting a right or left ventricular (LV) outflow tract origin (i.e., left bundle branch block (LBBB)-like with a vertical axis pattern) or from the posterior fascicle of the left bundle branch (i.e., narrow QRS (<130 ms) with a right bundle branch block (RBBB)-like pattern and superior axis). These PVBs usually decrease during effort [61,62].

PVBs with an LBBB-like morphology and intermediate or superior axis (denoting a right ventricular (RV) or septal origin), and those with an RBBB-like morphology, wide QRS and superior axis (indicating an LV lateral wall origin), are classified as “uncommon” because they carry a higher risk of being associated with an underlying disease. The suspect is reinforced when arrhythmias increase in number or complexity during effort [16,65,66,67,68,69].

The traditional concept that arrhythmic risk is predicted by “PVB burden” (i.e., numerous PVBs are associated with a higher probability of malignant arrhythmic events) has been disproved by recent evidence. Indeed, it has been demonstrated that benign extrasystolic foci (more often located in the outflow tract) can give rise to a very high number of PVBs on 24 h Holter ECG in the absence of a pathological substrate [61,62]. Several studies have demonstrated the benign prognosis of PVBs originating from ventricular outflow tracts, even when numerous [8,62,69], with the exception of the rare complication of tachycardia-induced LV dysfunction, the so-called “tachycardiomyopathy” [8,69,70,71]. Recent evidence suggests an inverse correlation between the number of PVBs and the probability of detecting a NLVS in patients with VAs, both in the general population [72] and athletes [73]. Consequently, the number of PVBs on ambulatory ECG monitoring is no longer deemed the main indication for further diagnostic testing to exclude a cardiomyopathy [9].

The evaluation of athletes with PVBs should take into consideration other clinical data: in particular, a positive family history of SCD or cardiomyopathy, history of unexplained syncope or the presence of other major ECG abnormalities (such as T wave inversion) on resting ECG.

## 5. Relationship between PVB Features and NLVS Revealed by CMR

In the last year, CMR imaging has been increasingly prescribed to investigate athletes with PVBs even when other clinical investigations (particularly echocardiography) are normal. This examination has led to the discovery of isolated NLVSs as the origin of PVBs in some cases that would otherwise have been considered idiopathic.

The first systematic study on athletes with VAs in association with an isolated NLVS was reported by Zorzi et al., in 2016 [25], although a small case series was previously described [74]. In this study, a large majority of the patients underwent CMR imaging for apparently idiopathic VAs, while only a minority did so because of ECG alterations (lateral T wave inversion and/or low QRS voltages) or echocardiographic abnormalities (lateral wall hypokinesia). In most cases, the PVB morphology was RBBB-like, wide QRS and superior axis, consistent with an origin from the LV lateral free wall. Significantly, during a mean follow-up of just over 3 years, major arrhythmic events such as appropriate defibrillator shock, sustained VT or SCD occurred in six (22%) patients, mostly during sport activity. This finding underlines the role of adrenergic stimulation as an arrhythmic trigger in NLVSs. Recent evidence confirmed the association between NLVSs and VAs in athletes. A recent study by Lie et al. compared 43 high-performance competitive athletes with VAs to healthy athletes using clinical data and cardiac imaging and found a higher prevalence of non-ischemic LGE on CMR in athletes with VAs compared to healthy controls [75].

Several studies have demonstrated that PVB morphology and behavior during exercise can predict the presence of an underlying NLVS in athletes with apparently idiopathic VAs. Cipriani et al. reported CMR findings in a series of athletes with PVBs who were referred to a tertiary center [22]. Finding a myocardial scar (LGE) was three times more probable (47% vs. 17%) if PVBs persisted or worsened during a stress test compared to those with a reduction in VAs at increasing workload. The other independent predictors of abnormal CMR were an RBBB-like morphology of PVBs and T wave inversion on resting ECG (Figure 3).

Another investigation aimed to evaluate the PVB burden in apparently healthy young athletes who volunteered to undergo a 12-lead 24 h Holter ECG, including a training session. Seventeen subjects were subsequently referred to CMR for frequent (>500/day), repetitive or training-induced PVBs [73]. In three cases, CMR demonstrated a NLVS: in all three, the PVBs showed an RBBB-like morphology with an inferior axis, and they were triggered by the exercise (Figure 4).

In a multicenter study reporting on 251 competitive athletes with a negative family history and unremarkable ECG and normal echocardiography findings who underwent CMR for evaluation of VAs, LV LGE was found in 28 (11%). The scar often showed a subepicardial/midmyocardial distribution (non-ischemic) and was more likely when PVBs depicted an RBBB-like pattern or multiple morphologies, or when non-sustained VT was recorded during exercise testing. Conversely, a high number of PVBs (>3300/day) predicted a normal CMR (Figure 5) [23].

The systematic evaluation of PVBs occurring during exercise testing increases the diagnostic sensitivity of preparticipation screening compared to traditional first-line investigations (history, physical examination and 12-lead ECG). In a population of 10,985 young athletes, adding exercise testing for VAs led to a 75% increase in the diagnosis of pathological cardiac substrates, mainly NLVSs (Figure 6). The morphology of PVBs (RBBB-like or both RBBB and LBBB) and the occurrence of repetitive VAs during exercise correlated with an underlying NLVS at CMR [24].

Finally, the correlation between the characteristics of PVBs and the probability of an isolated NLVS as well as the prognostic significance of this CMR finding has also been documented in the general population. In a study by Muser et al. on 518 subjects with frequent PVBs, LGE was diagnosed by CMR in 16%. Polymorphic PVBs and PVBs with a non-infundibular morphology were predictors of pathological myocardial substrates. Nearly one third of the subjects with LGE suffered major arrhythmic events during a median follow-up of 67 months (three died suddenly during effort), underscoring, once again, the precipitating role of adrenergic stimulation as an arrhythmic trigger in patients with a NLVS [76]. The same authors subsequently reported that the ring-like pattern of NLVSs, defined as subepicardial/midmyocardial LGE involving at least three contiguous LV segments in the same short-axis slice, was associated with a particularly ominous outcome [77].

In summary, NLVSs are a relevant substrate for malignant VAs, mostly for those occurring during sport activity. The two main PVB characteristics that should trigger CMR prescription to rule out an underlying scar in athletes with otherwise normal clinical findings are the morphology (particularly RBBB-like, wide QRS and superior axis) and the increase in number and complexity during exercise.

## 6. Diagnostic Work-Up of PVBs in Athletes: When to Prescribe CMR

In athletes with PVBs, a careful clinical examination is required to rule out an underlying myocardial disease that could potentially cause SCD. Although more recent recommendations suggest performing additional testing only when two or more PVBs are recorded on the resting ECG, we believe that even a single “uncommon” PVB should be considered as a red flag [78].

First-line investigations in athletes with PVBs should include history, physical examination, exercise testing, 24 h ambulatory ECG including a training session (possibly with a 12-lead system to allow PVBs morphology assessment) and echocardiography [9]. The presence of an abnormal resting ECG in athletes with PVBs significantly increases the probability of myocardial disease. The main ECG abnormalities linked to cardiomyopathies or channelopathies are repolarization abnormalities (T wave inversion and ST segment depression), long QT, short QT, the Brugada pattern, conduction disturbances, ventricular pre-excitation and pathological Q waves [60,79]. In addition, signal-averaged ECG, which records slow conduction in the myocardial area that can become a substrate for monomorphic VTs, may play a role in the diagnostic process [80,81].

Exercise testing is a key tool for unmasking electrocardiographic abnormalities or arrhythmias absent at baseline. The effort should be maximal, and the test should be carried out until muscular exhaustion, not just limited to 85% of the predicted maximal heart rate. This is important not only for the documentation of ischemic ST-T changes, but also for eliciting adrenergic-dependent PVBs that may occur only at the peak of effort. Echocardiography is essential to rule out valvular defects, cardiomyopathies, congenital disorders and the anomalous origin of coronary arteries [82,83].

In case one of the above investigations is abnormal, additional tests may be needed based on clinical suspicion (e.g., CMR to rule out a possible cardiomyopathy, or a coronary computed tomography angiogram to exclude an ischemic cardiomyopathy). If first-line investigations are normal, indications for CMR depend on the characteristics of the PVBs (morphology, complexity and relation to exercise) that have been discussed above. Figure 7 shows a proposal of the diagnostic work-up of athletes with PVBs [9].

## 7. Potential Pitfalls in Interpretation of CMR in Athletes

When analyzing CMR imaging, attention must be paid to some pitfalls. It is known that CMR shows systematically larger chamber dimensions and volumes, but a smaller wall thickness and mass, compared with echocardiography [84]; thus, caution should be exercised in applying echocardiographic-derived reference values when performing CMR in athletes [31]. The presence of LGE should be confirmed in two orthogonal planes, and care must be taken in using the correct time-for-inversion (TI) to obtain the maximal contrast between normal myocardium and scar tissue; moreover, bright ghosting artifacts can result from poor ECG gating or poor breath holding. In addition, it is conventionally considered problematic to detect LGE at the level of the thin RV wall, even though newer generation CMR machines enhance the ability to identify RV intramyocardial scar tissue.

Not all non-ischemic LGE patterns should be considered markers of disease. Isolated right ventricular insertion point (or “junctional”) LGE, i.e., LGE confined to the insertion points of the right ventricle to the anterior and/or posterior ventricular septum, has been described in many pathological conditions such as pulmonary hypertension [85] or hypertrophic cardiomyopathy [86], but it is also commonly found in healthy athletes, especially those engaged in endurance disciplines [87,88,89]. In this subset, its prevalence reaches up to 30%, and it may reflect the tension on insertion points due to pressure and volume overload on the right ventricle during intensive exercise [49]. In subjects without additional evidence of cardiac disease, it should not be considered pathological [90]. Finally, a mid-wall stria of LGE in the basal portion of the interventricular septum may reflect a septal perforator coronary artery, rather than myocardial fibrosis; this must be taken into account in order to prevent cardiac disease overdiagnosis [91].

## 8. Recommendations for Competitive Sport Eligibility in Athletes with NLVSs

In both the European Society of Cardiology and the American Heart Association guidelines on competitive athletes, NLVSs are addressed in the myocarditis chapter [92,93]. The European document recognizes that myocardial scars can be the source of life-threatening VAs. Accordingly, competitive sport activity is not recommended in athletes with a NLVS associated with LV dysfunction or clinically relevant arrhythmias (frequent or complex PVBs). At present, there is no evidence that isolated LGE is associated with an increased risk of SCD during exercise, but athletes should at least remain under clinical surveillance [93]. Similarly, the American Heart Association’s recommendations suggest restricting sport activity in athletes with a NLVS and ventricular dysfunction or relevant arrhythmias. They also state the following: “at present, it is unresolved whether resolution of myocarditis-related LGE should be required to permit return to competitive sports” [93].

## 9. Conclusions

Systematic investigations of athletes with PVBs through CMR imaging in recent years have served to clarify that substrates of SCD, particularly NLVSs, may be missed in routine clinical tests. In the presence of at-risk PVB characteristics, based on their morphology, complexity and relation to effort, CMR should be performed even when echocardiography and baseline ECG are normal. Besides standard CMR scans and post-contrast images, novel techniques such as mapping and feature tracking are also emerging in the subset of athletes’ evaluation.

Beyond NLVS, ischemic heart disease represents an important substrate of VAs, especially in master athletes. In the evaluation of older athletes with PVBs, stress CMR in addition to the standard CMR protocol might be useful for ruling out coronary artery stenosis, although further studies are needed to evaluate its diagnostic value in this peculiar clinical setting.

## Figures and Tables

**Figure 1 jcm-11-00426-f001:**
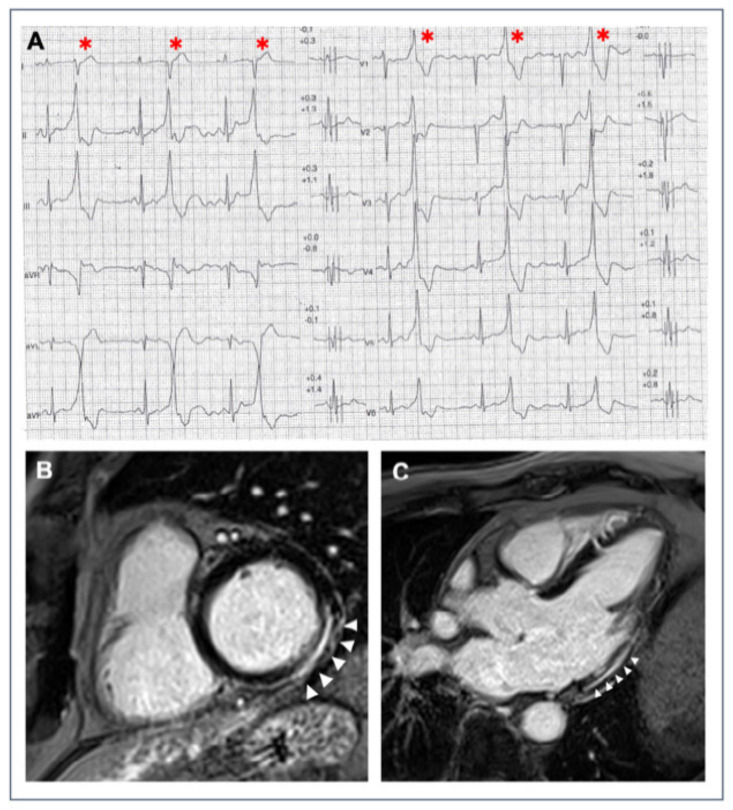
A 40-year-old athlete presented with frequent PVBs on ECG ambulatory monitoring. Exercise test revealed frequent and repetitive monomorphic PVBs with RBBB/inferior axis morphology ((**A**), red asterisks). Post-contrast sequences on CMR showed an LGE subepicardial/midmyocardial stria in the basal inferolateral and inferior LV walls (white triangles; (**B**) short-axis view; (**C**) 3-chamber view).

**Figure 2 jcm-11-00426-f002:**
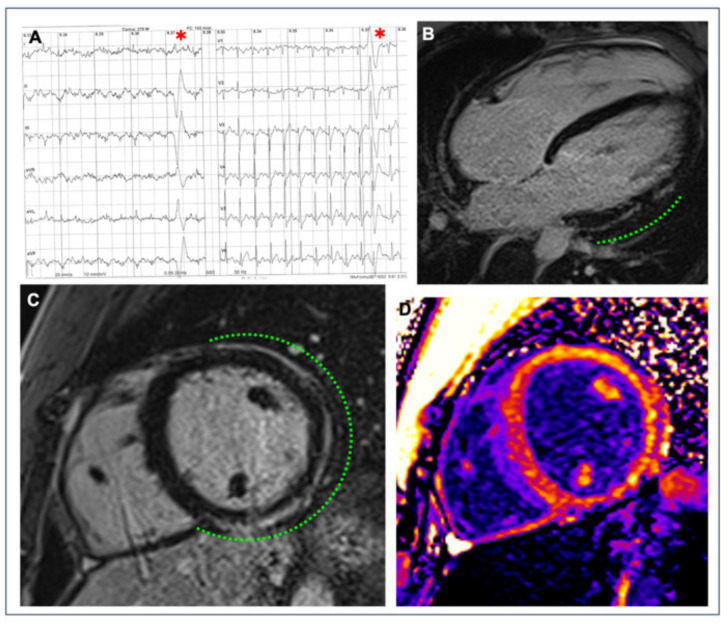
A 26-year-old competitive hockey player presented with frequent PVBs with right bundle branch block/superior axis morphology at high workload during exercise testing ((**A**), red asterisks). Post-contrast sequences on CMR revealed a subepicardial stria of LGE with a “ring-like” pattern, involving the anterior, lateral and inferior LV walls in their basal and medium portions (green dotted line; (**B**) 4-chamber view; (**C**) short-axis view); the presence of fibrous tissue is also confirmed by the increased signal in the correspondent areas in the native T1 mapping short-axis sequence (**D**).

**Figure 3 jcm-11-00426-f003:**
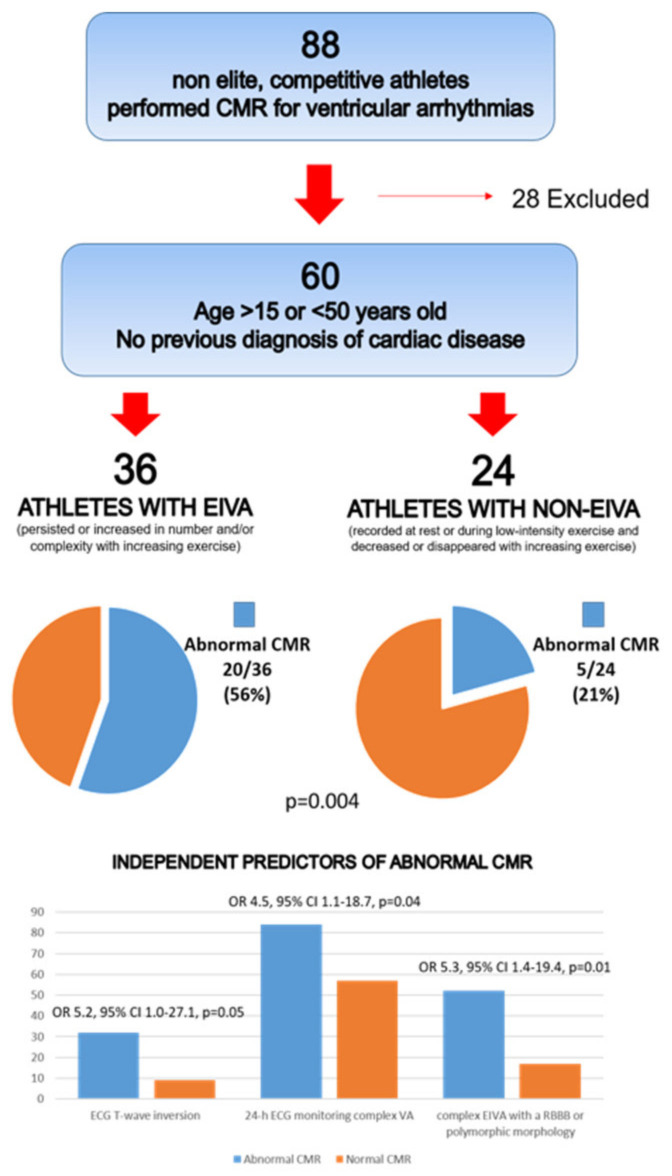
CMR: cardiac magnetic resonance; EIVA: exercise-induced ventricular arrhythmias; ECG: electrocardiogram; VA: ventricular arrhythmias; RBBB: right bundle branch block. Reproduced with permission from [22].

**Figure 4 jcm-11-00426-f004:**
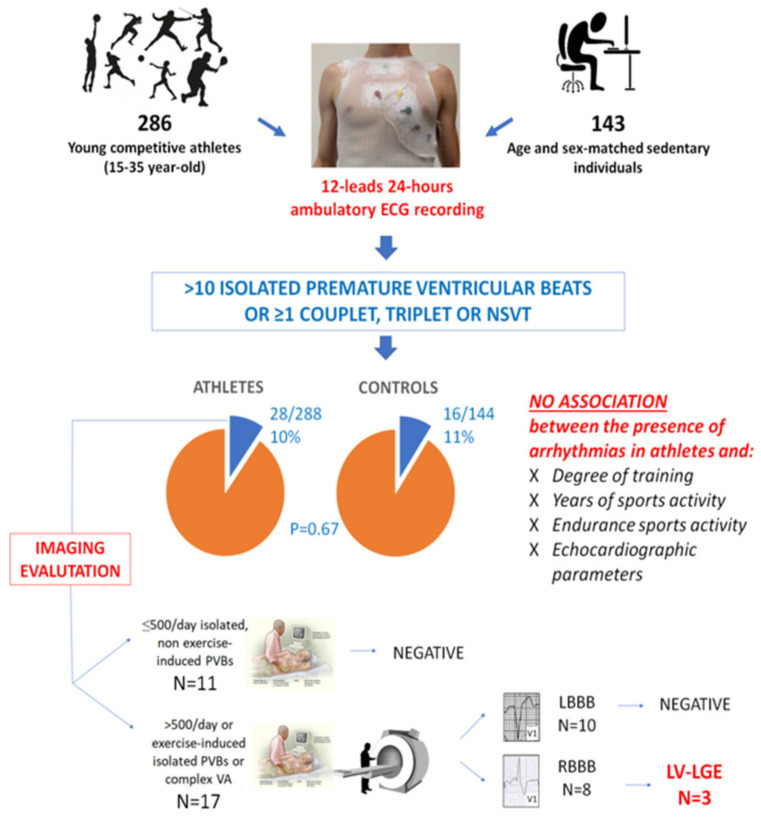
Summary of the main findings of a study enrolling 286 young competitive athletes and 143 healthy sedentary controls who volunteered to undergo 12-lead 24 h ambulatory ECG monitoring. LBBB, left bundle branch block; LGE, late gadolinium enhancement; LV, left ventricular; NSVT, non-sustained ventricular tachycardia; PVB, premature ventricular beat; RBBB, right bundle branch block; VA, ventricular arrhythmia. Reproduced with permission from [73].

**Figure 5 jcm-11-00426-f005:**
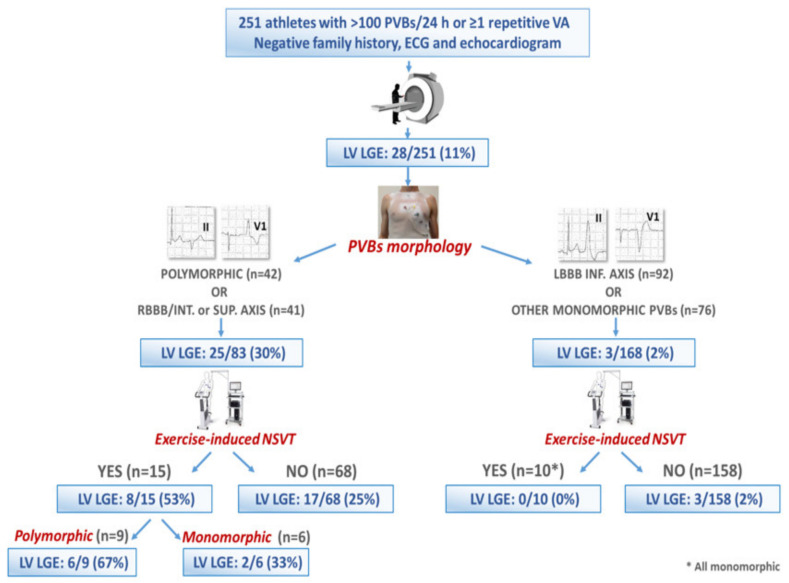
Summary of the main findings of a multicenter study on 251 athletes with ventricular arrhythmias, normal echocardiography and unremarkable ECG findings and a negative family history. Reproduced with permission from [23].

**Figure 6 jcm-11-00426-f006:**
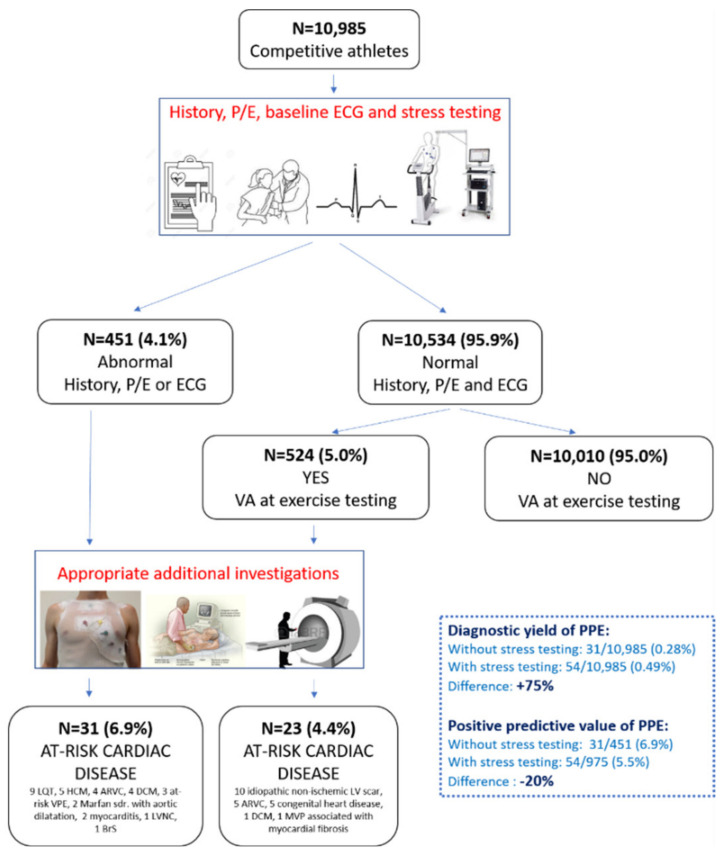
Summary of the main findings of a study investigating the additional value of exercise testing for ventricular arrhythmias in addition to history, physical examination and ECG in the setting of athletes’ preparticipation screening. Reproduced with permission from [24].

**Figure 7 jcm-11-00426-f007:**
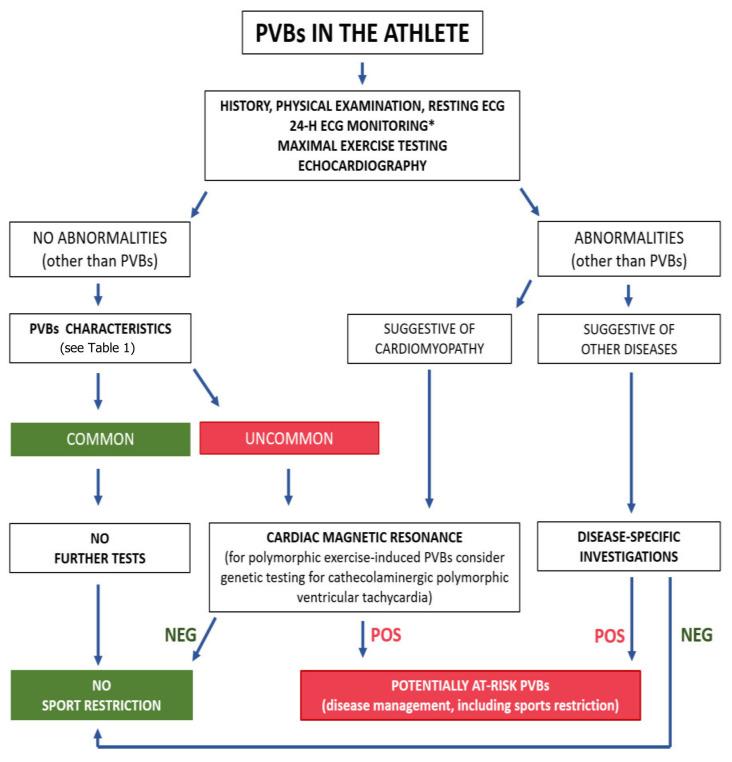
Proposed algorithm for evaluation of athletes with premature ventricular beats. * Twenty-four-hour ECG monitoring should ideally have a 12-lead configuration and include a training session. NEG, negative; POS, positive; PVBs, premature ventricular beats. Reproduced with permission from [9].

**Table 1 jcm-11-00426-t001:** Classification of PVBs morphology according to the probability of an underlying pathological myocardial substrate.

QRS Morphology	Probable Origin of PVB	Disease Probability	V1 Pattern	aVF Pattern	Refs.
**Common**					
LBBB, late precordial transition (R/S = 1 after V3), inferior axis.	Right ventricular outflow tract.	Usually benign.	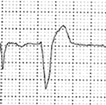	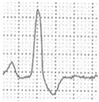	[8,61,62]
LBBB, inferior axis, small R waves in V1, early precordial transition (R/S = 1 by V2 or V3).	Left ventricular outflow tract.	Usually benign.	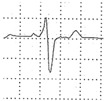	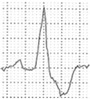	
QRS <130 ms resembling a typical RBBB/left anterior fascicular block.	Left posterior fascicle of the left bundle branch.	Usually benign.	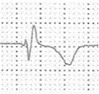	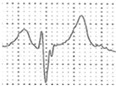	[63]
QRS <130 ms resembling a typical RBBB/left posterior fascicular block.	Left anterior fascicle of the left bundle branch.	Usually benign.	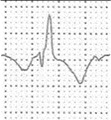	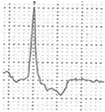	
**Uncommon**					
Atypical RBBB, QRS ≥130 ms, positive QRS in V1–V6 and inferior axis.	Anterior mitral anulus/left ventricular outflow tract.	Usually benign but may be associated with myocardial disease.	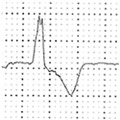	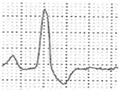	[64]
Atypical RBBB, QRS ≥130 ms, intermediate or superior axis.	Left ventricular free wall.	May be associated with myocardial disease.	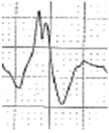	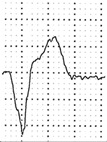	[65,66]
LBBB, superior or intermediate axis.	Right ventricular free wall or interventricular septum.	May be associated with myocardial disease.	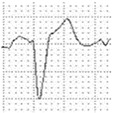	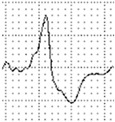	

LBBB = left bundle branch block, i.e. negative QRS complex in V1; RBBB = right bundle branch block, i.e. positive or isodiphasic QRS complex in V1.

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
