# Peer review of "Role of Cardiac Magnetic Resonance Imaging in the Evaluation of Athletes with Premature Ventricular Beats"

_jcm, 2022, doi:10.3390/jcm11020426_

Round 1

Reviewer 1 Report

In this narrative review, Dr. Brunetti and colleagues discussed the current clinical diagnosis of premature ventricular beats (PVBs) in athletes and the role of CMR to unravel the 'unseen'. Overall, this is a nicely written manuscript, with a high readability. I also believe that this manuscript has a strong clinical insights. Yet, I do have some minor suggestions to be incorporated:

  • I would suggest to highlight the clinical perspective of this paper by changing the title into "Clinical evaluation of premature ventricular beats in athletes: focus on the application of Cardiac Magnetic Resonance imaging"
  • The aim of the review needs to be specified in the abstract. Consider adding to clarify the focus of this paper.
  • Line 26: please remove "just".
  • Line 26: I think the authors could change "arrhythmic cardiac arrest" into "life-threatening cardiac arrhythmias" because cardiac arrest is always caused by arrhythmia (VT/VF/Asystole/PEA).
  • Line 48: please add hyphen in "T-wave inversion". The authors frequently missed hyphen in the text so please recheck everything once again. 
  • I would suggest to add the supporting references / citations as a column in Table 1, especially indicating which studies reported the benign classifications of common ECG patterns and so on. I know that the authors have included the citation in the paragraphs below the table but it would be clearer to put it next to each row in the table. 
  • Also, if possible, I would suggest to add representative examples of the PVBs in Table 1. The authors could only include relevant ECG leads. Having this visualization would help the reader to quickly 'catch' the mentioned ECG patterns.
  • The authors said "Recent studies have questioned the traditional concept that the arrhythmic risk is proportional to the “PVBs burden”...Consequently, the number of PVBs on the ambulatory ECG Holter monitoring is no longer deemed an indication to further diagnostic testing to exclude a cardiomyopathy". I am wondering if this is a valid statement because having more frequent PVCs would increase the probability of getting R-on-T and TdP, which will increase the arrhythmic risk? Please comment on this.
  • Line 78: Please define SCD in its first use.
  • Line 87: "increased calcium inward current in the myocyte following adrenergic stimulation," This is not entirely true as in CPVT, the major issue is not the calcium inward current due to L-type Ca channel but because of the mutation in RyR2 or CASQ2, which increases the SR Ca release following adrenergic stimulation. The authors could benefit from this review (PMID: 32188566).
  • Although it is not within the scope of this review, maybe the authors could say 1-2 sentences about the possible role of PET scan with FDG or oxygen tracers? How inferior or superior it is compared to CMR?
  • Before conclusion, it would be useful to know some future directions of this topic, within and beyond CMR. Is there anything to improve the early diagnosis of CV diseases in athletes to prevent SCD? What is next on the research in the SCD prevention in athletes? etc.

Author Response

  1. In this narrative review, Dr. Brunetti and colleagues discussed the current clinical diagnosis of premature ventricular beats (PVBs) in athletes and the role of CMR to unravel the 'unseen'. Overall, this is a nicely written manuscript, with a high readability. I also believe that this manuscript has a strong clinical insights. Yet, I do have some minor suggestions to be incorporated: I would suggest to highlight the clinical perspective of this paper by changing the title into "Clinical evaluation of premature ventricular beats in athletes: focus on the application of Cardiac Magnetic Resonance imaging".

R: we thank the Revisor for the suggestion that emphasizes the clinical implication of the article; according to His/Her comment, we modified the title as suggested.

  1. The aim of the review needs to be specified in the abstract. Consider adding to clarify the focus of this paper.

R: thanks to Reviewer suggestion, we added the aim of the review in the abstract (lines 21-22).

  1. Line 26: please remove "just"; Line 26: I think the authors could change "arrhythmic cardiac arrest" into "life-threatening cardiac arrhythmias" because cardiac arrest is always caused by arrhythmia (VT/VF/Asystole/PEA). Line 48: please add hyphen in "T-wave inversion". The authors frequently missed hyphen in the text so please recheck everything once again.

R: we thank the Reviewer for these suggestions, and we provided to modify the text as indicated (lines 29-30-51-89-133-191).

  1. I would suggest to add the supporting references/ citations as a column in Table 1, especially indicating which studies reported the benign classifications of common ECG patterns and so on. I know that the authors have included the citation in the paragraphs below the table but it would be clearer to put it next to each row in the table. Also, if possible, I would suggest to add representative examples of the PVBs in Table 1. The authors could only include relevant ECG leads. Having this visualization would help the reader to quickly 'catch' the mentioned ECG patterns.

R: we thank the Reviewer because these are very useful suggestions that surely makes the table more easily interpretable; we added specific references to Table 1, as well as examples of PVBs morphology, on the table; for space issued, we provided only the representative morphologies in two leads, V1 and aVF.

  1. The authors said "Recent studies have questioned the traditional concept that the arrhythmic risk is proportional to the “PVBs burden”...Consequently, the number of PVBs on the ambulatory ECG Holter monitoring is no longer deemed an indication to further diagnostic testing to exclude a cardiomyopathy". I am wondering if this is a valid statement because having more frequent PVCs would increase the probability of getting R-on-T and TdP, which will increase the arrhythmic risk? Please comment on this.

R: we respectfully disagree with the Reviewer that an increased PVBs burden is associated to an higher probability of TdP triggered by R-on-T phenomena. “R-on-Ts” (i.e. PVB that arises at the peak of the T-wave) are the expression of a phase 2-reentry typical of conditions characterized by severe repolarization inhomogeneity within the ventricles such as channelopathies (LQTS), acute ischemia, electrolytic disturbances or drug toxicity. Instead, frequent PVBs in young people and athletes are usually caused by triggered activity of an ectopic focus (particularly in the RVOT/LVOT) that may organize in runs of TVNS or even sustained TV, but not typically cause TdP. Accordingly, several studies both in the general population and in athletes have confirmed that very frequent PVBs are uncommonly associated with an underlying substrate and are benign (Ventura R et al. Decennial follow-up in patients with recurrent tachycardia originating from the right ventricular outflow tract: electrophysiologic characteristics and response to treatment. Eur Heart J 2007;28:2338-45; Kennedy HL et al. Long-term follow-up of asymptomatic healthy subjects with frequent and complex ventricular ectopy. N Engl J Med 1985;312:193-7; Gaita F. et al. Long-term follow-up of right ventricular monomorphic extrasystoles. J Am Coll Cardiol. 2001 Aug;38(2):364-70; Delise P. et al. Long-term effect of continuing sports activity in competitive athletes with frequent ventricular premature complexes and apparently normal heart. Am J Cardiol 2013;112:1396-402; Ventura R, Steven D, Klemm HU, et al. Decennial follow-up in patients with recurrent tachycardia originating from the right ventricular outflow tract: electrophysiologic characteristics and response to treatment. Eur Heart J 2007;28:2338-45; Niwano S, Wakisaka Y, Niwano H, et al. Prognostic significance of frequent premature ventricular contractions originating from the ventricular outflow tract in patients with normal left ventricular function. Heart 2009;95:1230-7). As far as the left ventricular scar is concerned, it has been demonstrated that the number of PVBs in individuals with an apparently normal heart is inversely correlated with the probability of detecting a LV scar on CMR, both in the general population (Muser, D. et al., Risk Stratification of Patients With Apparently Idiopathic Premature Ventricular Contractions. JACC Clin. Elec-trophysiol. 2020;6(6):722-735) and athletes (Crescenzi, C. et al. Predictors of left ventricular scar using cardiac magnetic resonance in athletes with apparently idiopathic ventricu-lar arrhythmias. J. Am. Heart Assoc. 2021;10(1):e018206). Therefore, available scientific evidence demonstrates that it is the presence of an underlying myocardial substrate, rather than the number of PVBs, that conditions the prognosis. Moreover, in the evaluation of athletes with ventricular arrhythmias, one must keep in mind to consider the presence of familial history of sudden cardiac death and cardiomyopathies and the presence of major symptoms such as prolonged palpitations or syncope.

However, it is noteworthy that very frequent PVBs may uncommonly lead to a dyssynchrony-related myocardial disfunction (tachycardiomyopathy). We discussed further this point in the revised manuscript (lines 76-87).

  1. Line 78: Please define SCD in its first use.

R: we thank the Reviewer for this clarification, and we provided to modify the text (line 89).

  1. Line 87: "increased calcium inward current in the myocyte following adrenergic stimulation," This is not entirely true as in CPVT, the major issue is not the calcium inward current due to L-type Ca channel but because of the mutation in RyR2 or CASQ2, which increases the SR Ca release following adrenergic stimulation. The authors could benefit from this review (PMID: 32188566).

R: we thank the reviewer for the clarification, and we modified the text also according to the review suggested (line 99).

  1. Although it is not within the scope of this review, maybe the authors could say 1-2 sentences about the possible role of PET scan with FDG or oxygen tracers? How inferior or superior it is compared to CMR?

R: as suggested by the Reviewer, we recognized the possible utility of PET imaging (especially for inflammatory heart diseases such as cardiac sarcoidosis) in the text (lines 158-165).

  1. Before conclusion, it would be useful to know some future directions of this topic, within and beyond CMR. Is there anything to improve the early diagnosis of CV diseases in athletes to prevent SCD? What is next on the research in the SCD prevention in athletes? etc

R: according to the Reviewer’s suggestion, we expanded the Conclusions section discussing an important future research direction, i.e. early identification of coronary artery disease masters athletes (lines 339-342, also further discussed in lines 104-107). Ischemic heart disease represents a major substrate of arrhythmias and is the main substrate of sudden cardiac death in older athletes. Hence, stress-CMR might be useful for ruling out coronary artery disease in older athletes with ventricular arrhythmias, although further studies are needed to confirm this hypothesis.

Reviewer 2 Report

I read with interest the review article entitled “Evaluation of the athlete with premature ventricular beats: focus on Cardiac Magnetic Resonance” by Brunetti et al.

The authors discuss the not uncommon and interesting topic that is the significance and management of premature ventricular beats (PVBs) in athletes.

The authors focus on family history, ECG characteristics including PVBs morphology and findings during exercise. Hence, the authors emphasize on detecting the athletes that require further testing with CMR.

I believe that on the top of what the authors already suggest, the diagnostic role and prognostic significance of signal averaged ECG (SAECG) should be noted (Gatzoulis et al, J. Arrhythmia, 2018). SAECG can predict the risk of arrhythmic sudden cardiac death in ischemic and dilated cardiomyopathy (Gatzoulis et al, Ann CardiolVasc Med. 2020)

Moreover, since the review is a focus on CMR I believe that the authors should emphasize more on the management of patients with PVBs and abnormal CMR.

Which CMR findings have been correlated with sudden cardiac death?

Which of those findings can be prevented with sports restriction?

When, based on previous and CMR findings, a electrophysiological study is warranted to predict SCD risk?

Minor:

Line 175 VAS->Vas

Line 178 are predict ->  can predict

Author Response

  1. I read with interest the review article entitled “Evaluation of the athlete with premature ventricular beats: focus on Cardiac Magnetic Resonance” by Brunetti et al. The authors discuss the not uncommon and interesting topic that is the significance and management of premature ventricular beats (PVBs) in athletes. The authors focus on family history, ECG characteristics including PVBs morphology and findings during exercise. Hence, the authors emphasize on detecting the athletes that require further testing with CMR. I believe that on the top of what the authors already suggest, the diagnostic role and prognostic significance of signal averaged ECG (SAECG) should be noted (Gatzoulis et al, J. Arrhythmia, 2018). SAECG can predict the risk of arrhythmic sudden cardiac death in ischemic and dilated cardiomyopathy (Gatzoulis et al, Ann CardiolVasc Med. 2020)

R: we thank the reviwer for the useful suggestion. Even if SAECG has a limited diagnostic accuracy in distinguishing specific diseases, it could be a useful tool in recording slow conduction myocardial area that may be a substrate for monomorphic VTs. We discussed this topic in the revised manuscript (lines 287-289) and cited the two relevant papers.

  1. Moreover, since the review is a focus on CMR I believe that the authors should emphasize more on the management of patients with PVBs and abnormal CMR. Which CMR findings have been correlated with sudden cardiac death? Which of those findings can be prevented with sports restriction? When, based on previous and CMR findings, a electrophysiological study is warranted to predict SCD risk?

R: management of athletes with PVBs related to an underlying pathological substrate is a very relevant but also very broad topic, as PVBs can be linked to a number of different diseases all with specific guidelines regarding risk-stratification, treatment and eligibility to competitive sports. However, as our review mainly focuses on “isolated non-ischemic left ventricular scar”, in the revised manuscript we discussed current recommendations on sports eligibility in athletes with ventricular arrhythmias and NILVS.

  1. Minor: Line 175 VAS->Vas. Line 178 are predict ->  can predict

R: we thank the Reviewer for this clarification, and we provided to modify the text (line 201 and line 204).

Round 2

Reviewer 2 Report

I thank the authors for the changes

Please note spelling mistakes and correct during proofreading e.g line 80 10000/die!!